# Exploratory meta-analysis of hypoxic transcriptomes using a precise transcript reference sequence set

Yoko Ono[1] , Hidemasa Bono[1,2]

**Gene expression studies are intrinsically biased, with many studies influenced by concomitant information such as gene–disease associations. This limitation can be overcome using a data-driven analysis approach without relying on ancillary information. The FANTOM CAGE–Associated Transcriptome project provides a comprehensive meta-assembly of the human transcriptome using coding and noncoding genes. Hypoxia strongly influences gene expression; in addition, noncoding RNA (ncRNA) metabolism is down-regulated in response to hypoxic stimuli. We evaluated the differential response of various transcripts to hypoxia by determining their hypoxia responsiveness scores. Enrichment analysis revealed that several genes associated with ncRNA metabolism, particularly those involved in ribosomal RNA processing, were down-regulated in response to hypoxia. Previously published information from the FANTOM CAGE–Associated Transcriptome project was suitable for meta-analysis of the transcriptome sequencing data from both coding and ncRNAs and to evaluate the hypoxia responsiveness of target transcripts and relationship between sense–antisense transcripts from the same locus. Our results may facilitate functional annotation of various transcripts including ncRNAs, allowing for both sense and antisense and coding and non-coding evaluations.**

## Introduction

Genomic research is inherently biased as it is heavily influenced by the known associations between specific genes and their families and the disease of interest (Mihai et al, 2021). Meanwhile, growing evidence from global molecular biology labs has facilitated the comprehensive functional annotation of much of the genome. Therefore, further development requires moving away from those familiar genes into the unknown using a data-driven approach. Our previous study revealed a clear publication bias regarding gene expression variation in response to hypoxia, and we identified a specific group of genes that had not been investigated in previous reports (Ono & Bono, 2021).

Oxygen is essential for the maintenance of vital functions, and cells have strict molecular mechanisms designed to help them cope with hypoxia. The study of these responses was dramatically advanced by the discovery of hypoxia-inducible factor-1 (HIF-1) in the 1990s (Semenza & Wang, 1992; Wang et al, 1995; Wang & Semenza, 1995). Under normoxic conditions, HIF is hydroxylated by prolyl hydroxylase and factor inhibiting HIF-1 (FIH-1) and then degraded by the ubiquitin–proteasome system to inhibit transcriptional activation (Jaakkola et al, 2001; Mahon et al, 2001). However, hypoxia reduces prolyl hydroxylase and FIH-1 activity, allowing HIF to escape its hydroxylation and promoting its interactions with the aryl hydrocarbon receptor nuclear translocator and transcriptional cofactor CREB, which in turn induces downstream gene expression (Ebert & Bunn, 1998). Our previous study revealed that genes related to the noncoding RNA (ncRNA) metabolic process were responsive to hypoxia (Ono & Bono, 2021).

Here, we evaluated the expression of genes related to ncRNA metabolism in response to hypoxia despite a lack of functional data describing their specific effects. In addition, although there is a balance in the data available for noncoding and coding RNAs, we believe that some changes in the expression of ncRNAs can only be revealed via data-driven analysis.

One tool facilitating these evaluations in the FANTOM5 project was designed to provide a reliable 5′ human lncRNA dataset for transcriptomic evaluations. These data were then combined with the data from the FANTOM5 Cap Analysis of Gene Expression (CAGE) project to produce the FANTOM CAGE–Associated Transcriptome (FANTOM-CAT) database. This dataset can be used to evaluate patterns in ncRNA expression using data from these and various coding genes. FANTOM CAGE data were used to identify ~28,000 lncRNA genes from the human genome, with ~20,000 of these transcripts likely to have some specific function (Hon et al, 2017).

Here, we used FANTOM-CAT as a reference and comprehensively scored the hypoxic response of various transcripts for both coding and noncoding genes. The data obtained in this study will help in the functional annotation of various transcripts, including ncRNAs, especially where there are fewer clues to their function.

---

[1]Laboratory of Genome Informatics, Graduate School of Integrated Sciences for Life, Hiroshima University, Higashihiroshima, Japan   [2]Laboratory of Bio-DX, Genome Editing Innovation Center, Hiroshima University, Higashihiroshima, Japan

Correspondence: bonohu@hiroshima-u.ac.jp

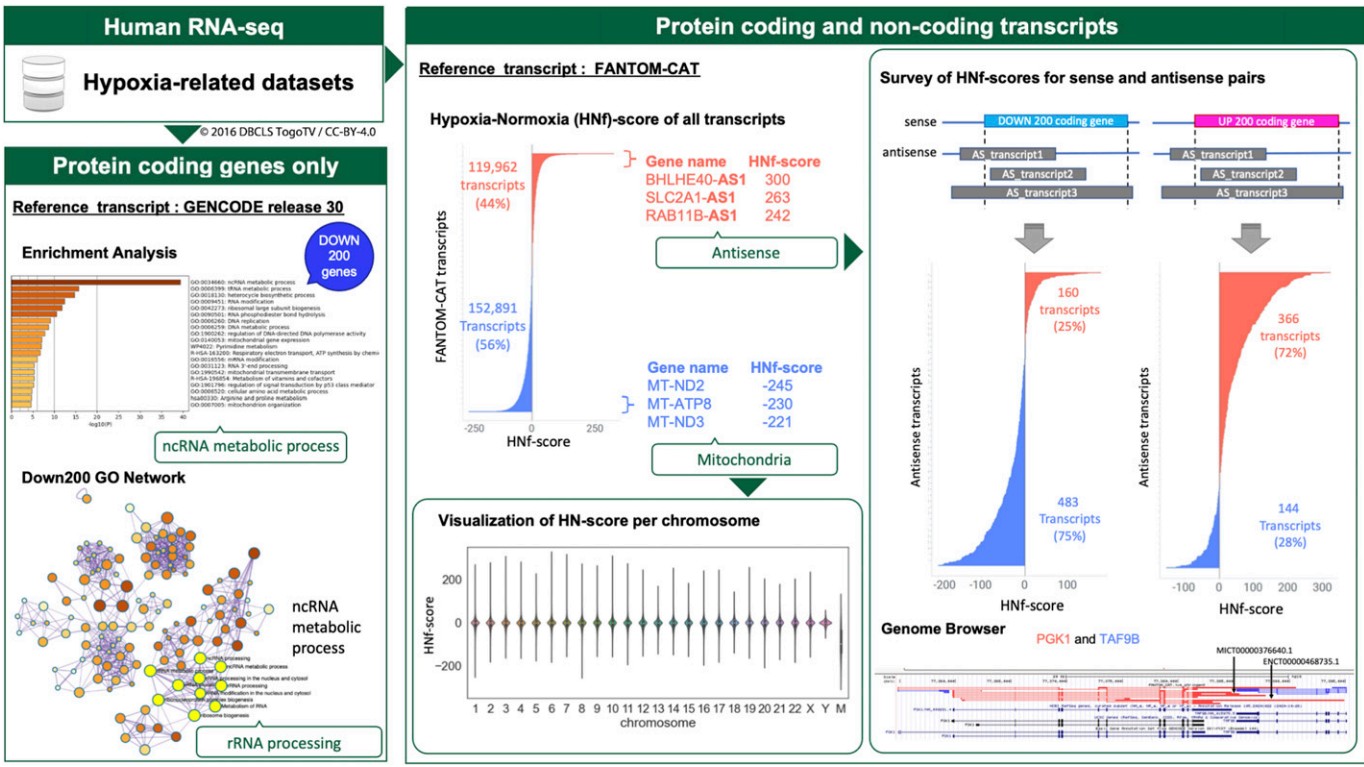

**Figure 1. Schematic overview describing our exploratory study of hypoxia-responsive transcripts.**
Data source: human hypoxia RNA-seq data from the NCBI Gene Expression Omnibus database. Protein-coding genes only: evaluated using GENCODE as the reference. Evaluation of the genes down-regulated by hypoxia revealed that they were enriched for transcripts involved in rRNA processing. Protein-coding and noncoding transcripts: evaluated using FANTOM CAGE–Associated Transcriptome as the reference. The number of transcripts derived from mitochondrial DNA and antisense analysis was reduced, suggesting a transcriptional regulation relationship in both sense and antisense genes (*PGK1* and *TAF9B*).

# Results

## Overview

The analyses were designed to evaluate hypoxia-responsive genes and transcripts based on two reference transcript sets (Fig 1).

In the analyses using two reference transcripts, one describes the quantitative gene expression data annotated using GENCODE release 30 and focuses only on protein-coding genes, and the other describes the expression data for both protein-coding and noncoding transcripts using FANTOM-CAT.

Initial evaluations of the coding-only dataset revealed that hypoxic stimulation suppressed the ncRNA metabolic process, especially genes involved in rRNA processing. Given this, we went on to expand the scope of our evaluations to include both protein-coding and noncoding transcripts and comprehensively evaluated the effects of hypoxia on both types of transcripts using a FANTOM-CAT dataset as a reference.

This dataset and its construction were described in detail in our previous study (Ono & Bono, 2021). The hypoxic conditions of the analyzed data were summarized for 495 samples (Fig 2A and B). Normoxic conditions were considered as a 20% oxygen concentration, although some samples were not mentioned, whereas hypoxic conditions ranged from 0.1 to 5% oxygen concentrations and some cases of hypoxia induced by chemicals such as $CoCl_2$. The

treatment time ranged from 1 h to 3 mo. The most common condition of hypoxia in the dataset was a 1% oxygen concentration for 24 h of treatment. Sixty-five percent of all samples were of cancer origin, with breast cancer as the most common tissue of origin. Gene expression in each tissue under representative conditions (cancer, oxygen concentration 1%, 24 h treatment) is shown as the $log_2$-transformed fold-change compared with under normoxic condition (Fig 2C).

## Down-regulation of ncRNA metabolism-related genes by the hypoxic response

From a sample of hypoxia-related datasets obtained from public databases, we selected sample pairs of hypoxia and normoxia pairs (HN-pairs). Based on these HN-pairs, plus 1 was calculated for each HN-pair if the expression variation was greater than 1.5-fold compared with normoxic sample and minus 1 if the variation was less than the reciprocal of 1.5-fold. These values were summed for each gene and designated as the hypoxia–normoxia score (HN-score). The HN-score qualitatively reflects hypoxia responsiveness, unlike the method used in general gene expression analysis. This HN-score was named as the HNg-score when the reference sequence was GENCODE and as the HNf-score when the sequence was based on FANTOM-CAT. We subjected 200 genes with high HNg-scores and 200 genes with low HNg-scores to enrichment analysis

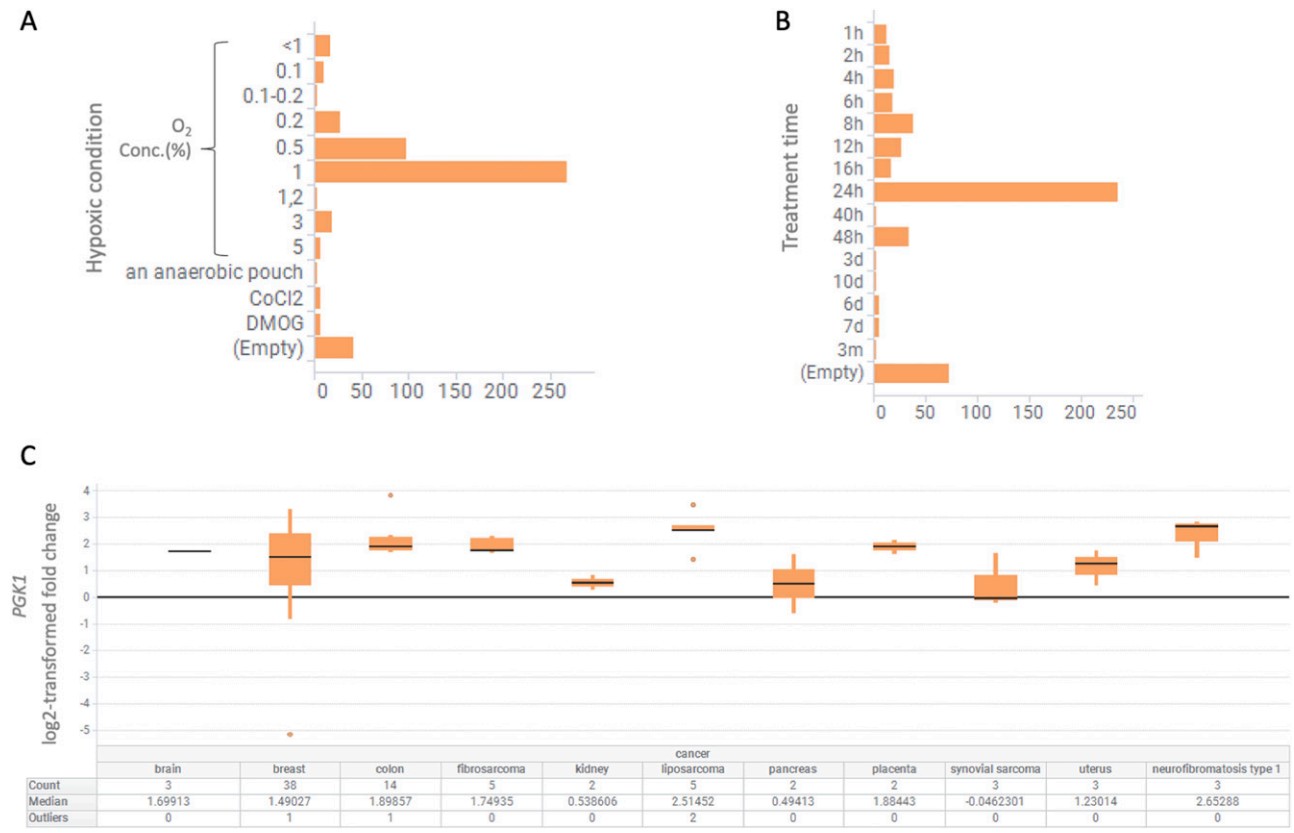

**Figure 2. Contents of hypoxia-related dataset.**
**(A, B)** The dataset is (A) summarized for oxygen concentration and (B) treatment time. **(C)** Gene expression information for *PGK1* derived from cancer cells treated for 24 h with a 1% oxygen concentration, the most representative condition, is displayed in a boxplot.

and evaluated whether the genes selected based on the HNg-score could be used to examine hypoxia responsiveness.

Enrichment analysis focusing only on protein-coding genes revealed that the top 200 genes identified by HNg-score, a measure of susceptibility to hypoxia, were enriched for hypoxia response–related gene sets (Fig 3A). However, enrichment analysis of the bottom 200 genes identified based on their HNg-score revealed suppression of expression of ncRNA metabolism–related genes (Fig 3B). Detailed investigation showed that most of these genes were associated with rRNA processing.

In enrichment analysis, the ncRNA metabolic process was further subdivided into the following Gene Ontology (GO) terms: ncRNA processing, ribosome biogenesis, ribonucleoprotein complex biogenesis, rRNA processing, and rRNA metabolic process (Fig 3C). The results of rRNA GO network analysis were visualized in UpSet plots, which showed that the ncRNA metabolic process is heavily reliant on most of these rRNA metabolic genes (Fig 3D).

### Survey of the number of articles referring FANTOM

FANTOM-CAT is a comprehensive catalog of human lncRNAs that improves upon existing lncRNA transcript models. The results obtained using this dataset have greatly contributed to life science research (Imada et al, 2020, 2021; Ramilowski et al, 2020). Despite the usefulness of this dataset, we predicted that it is used less

frequently for RNA-sequencing (RNA-seq) quantification compared with GENCODE; we quantitatively examined this hypothesis. If FANTOM-CAT is used less frequently, results obtained using this database may show a lower influence of publication bias (Ono & Bono, 2021), as demonstrated in our previous report. The number of articles in PubMed Central (PMC) that contained references to GENCODE, FANTOM, and RNA-seq was calculated for each year, with the number of RNA-seq articles exceeding 30,000 in 2021, whereas the number of articles referencing both FANTOM and RNA-seq was 0.5%, about one-tenth of the number referencing RNA-seq and GENCODE (Table 1).

### HNf-score evaluations of transcripts using FANTOM-CAT

The HNf-scores of all transcripts, including ncRNA, were quantified using FANTOM-CAT as a reference. We then went on to evaluate both the most increased and decreased transcripts using these values as a guide (Fig 4A). A total of 25 transcripts for both the UP and DOWN datasets, as delimited by HNf-score, are listed in Table 2, and a complete list of genes can be found within our archived figshare file (https://doi.org/10.6084/m9.figshare.19679493.v1).

This investigation revealed the fact that there were several cases where hypoxia-responsive transcripts with high HNf-scores were located within the antisense regions of genes with high HNg-scores (*BHLHE40-AS1*, *SLC2A1-AS1*). In addition, these evaluations

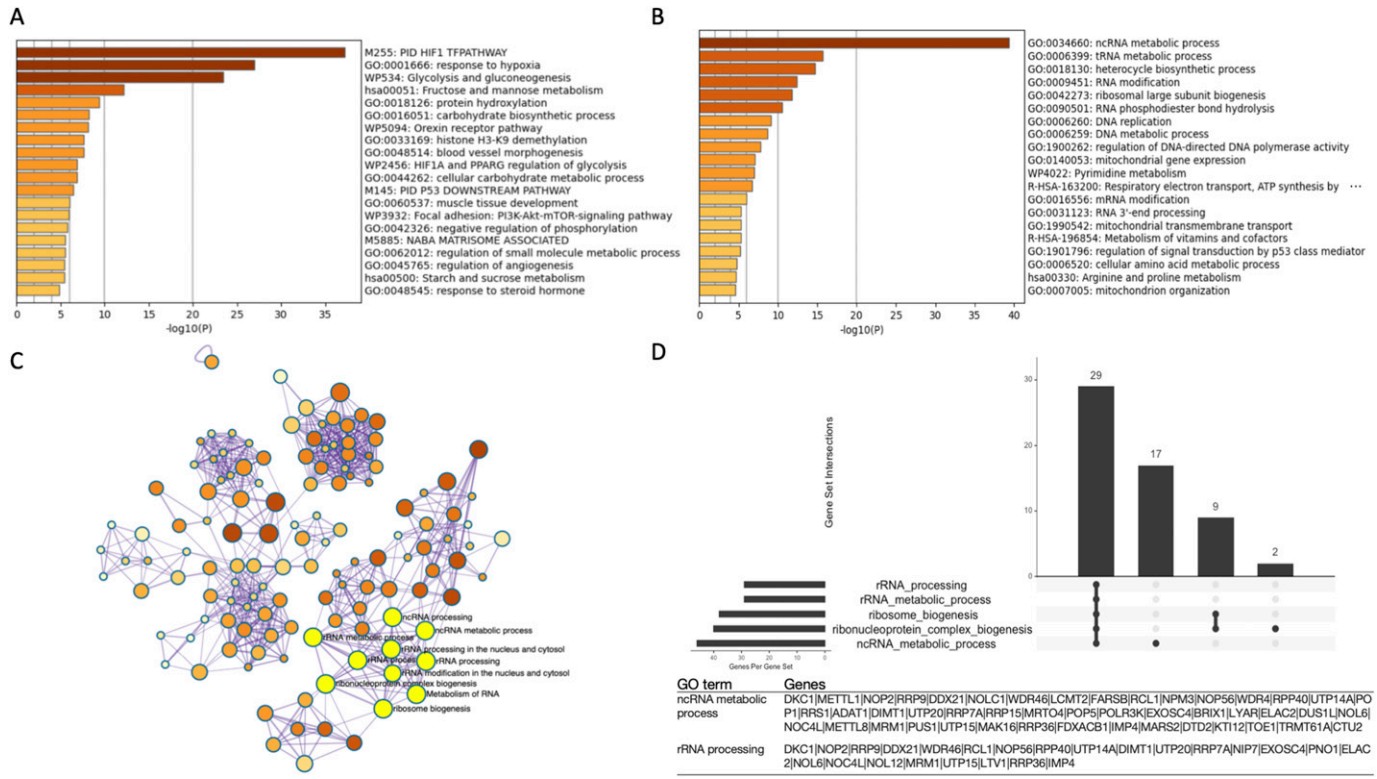

**Figure 3. Hypoxic stimulation suppresses the expression of noncoding RNA (ncRNA) metabolic process-related genes involved in rRNA processing.**
**(A, B)** Enrichment analysis for (A) the 200 UP regulated gene list and (B) the 200 DOWN regulated gene list. **(C, D)** Gene ontology (GO) network and (D) UpSet plot showing the number of genes included in GO in this analysis, describing the relative changes in ncRNA metabolism associated genes in the down-regulated gene list. In the GO network, the ncRNA metabolic process and rRNA processing focused on in this study are shown in yellow.

**Table 1. Number of reports describing GENCODE, FANTOM, and RNA-seq data.**

| Year | "RNA-seq" | "RNA-seq" and "GENCODE" (%) | "RNA-seq" and "FANTOM" (%) |
|------|-----------|------------------------------|------------------------------|
| 2017 | 17,373 | 769 (4.4%) | 123 (0.7%) |
| 2018 | 20,400 | 996 (4.9%) | 127 (0.6%) |
| 2019 | 25,668 | 1,254 (4.9%) | 130 (0.5%) |
| 2020 | 33,607 | 1,593 (4.7%) | 176 (0.5%) |
| 2021 | 40,757 | 1,976 (4.8%) | 210 (0.5%) |
| 2022 | 9,420 | 406 (4.3%) | 39 (0.4%) |

uncovered that most of the low HNf-score transcripts were associated with mitochondrial genes (*MT-ND2*, *MT-ATP8*). Hypoxic stimuli were shown to affect mitochondria. Visualization of the HNf-score for each chromosome showed that the median HNf-score was negative for mitochondria-derived transcripts (Fig 4B).

## Investigating the HNf-score of antisense transcripts

We investigated each antisense transcript individually as we could not perform enrichment analysis which requires information to comprehensively evaluate the effects of sense and antisense transcripts. The antisense transcripts for each of the UP200 and DOWN200 hypoxia-responsive coding genes were classified into one of four categories based on their gene expression behavior (Fig 5A and Table 3).

These evaluations revealed that ~75% of the antisense transcripts mirrored the behavior of their sense counterparts. However, there was still a small number of sense–antisense pairs presenting with opposite expression behaviors. This indicated that they were under different transcriptional control.

We also investigated transcripts with high or low HNf-scores to qualitatively evaluate hypoxia responsiveness. This evaluation also revealed that the top 10 antisense transcripts in the UP200 gene list were dominated by both short and long ncRNAs (lncRNAs), whereas the bottom 10 transcripts from this group were shown to be predominantly mRNA transcripts. Interestingly, among the antisense transcripts from the DOWN200 gene set, phosphoglycerate kinase 1

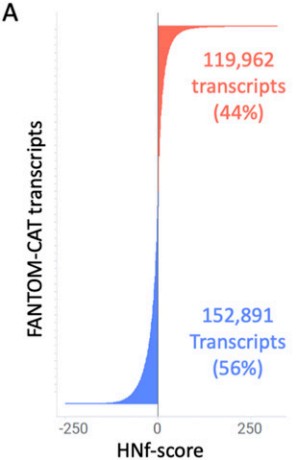

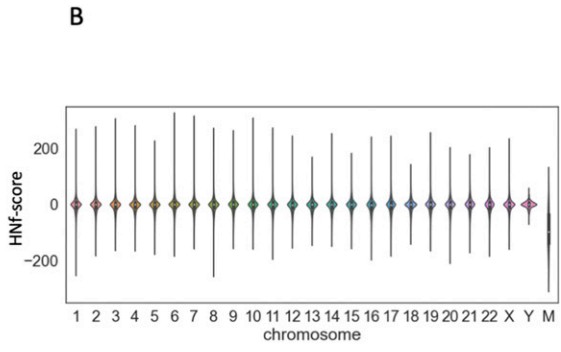

**Figure 4. Hypoxia–normoxia (HN) score of coding and noncoding genes' (HNf-score) evaluation of transcript using FANTOM CAGE–Associated Transcriptome.**
**(A)** HNf-score distribution of FANTOM CAGE–Associated Transcriptome transcripts. **(B)** Violin plot of the HNf-score of each chromosome.

(*PGK1*), which was also included in the UP200 gene set, had the highest HNf-score.

Given this, we homed in on *PGK1* and its antisense TATA-box–binding protein–associated factor 9b (*TAF9B*), evaluating their chromosomal localization using UCSC Genome Browser and HN-score at the gene level (Fig 5B and C). This analysis revealed that although these two genes are not described as having a sense–antisense relationship when evaluated using the UCSC or GENCODE datasets, FANTOM-CAT–based analysis suggests that these two transcripts are actually in a sense–antisense relationship. By using FANTOM-CAT as a reference, we were able to present for the first time the possibility that these two genes affect the regulation of expression.

## Discussion

Human gene research is commonly biased toward known pathogenic genes and pathways that are fairly well established (Stoeger & Amaral, 2022). However, given the development of novel data-driven tools, it is now possible to move beyond the targets identified in previous articles and into the realm of big data. Our previous studies focusing on coding genes have revealed a clear publication bias when evaluating gene expression changes in response to hypoxia and identified a novel set of largely uncharacterized hypoxia-responsive elements (Ono & Bono, 2021). Despite this, the details of the "ncRNA metabolic process"–related genes whose gene expression was suppressed in response to hypoxic conditions remain unclear.

Thus, we designed this study to further clarify the hypoxic response using a two-pronged approach. First, we completed a detailed investigation of the "ncRNA metabolic process" genes focused on only the coding genes in the dataset, and then we expanded this evaluation to any of the hypoxia-responsive genes annotated in the FANTOM-CAT database, which includes both coding and noncoding elements (Fig 1). All these data are available to everyone on figshare (https://doi.org/10.6084/m9.figshare.c.5971218.v2). We believe that these datasets and this analysis approach will help further elucidate the mechanisms regulating the hypoxic response.

We previously reported that the expression of several genes associated with the "ncRNA metabolic process" was down-regulated by hypoxic stimuli. However, the specific metabolic processes remain unknown. Given this, we started our evaluations by completing a Gene Ontology evaluation in an effort to identify the likely functions of these target transcripts. Subsequent enrichment analysis of the top and bottom 200 genes, identified using their HNg-score which was calculated using GENCODE, revealed that the top 200 genes were commonly associated with hypoxia response–related transcripts, whereas the bottom 200 genes were enriched for ncRNA metabolic processes (Fig 3A and B). Then, GO network analysis revealed a strong relationship between the ncRNA metabolic process and rRNA processing genes (Fig 3C and D) (https://www.ebi.ac.uk/QuickGO/GTerm?id=GO:0034660). This suggests that among the most common down-regulated transcripts, most targeted or formed part of the rRNA processing pathway.

rRNA is a critical part of the ribosome known to facilitate protein synthesis. rRNA has also been linked to the degradation of cytoplasmic lncRNAs after their interaction with the ribosomes (Carlevaro-Fita et al, 2016), and several other reports describe ncRNA-mediated silencing of the rRNA (Schmitz et al, 2010), all of which suggests an intimate relationship between these two sets of RNA mediators. However, given our previous focus on only the coding genes, the effects of hypoxia on the total transcript population, including the ncRNAs, remain unclear. Given this, we went on to calculate the more comprehensive HNf-score, which evaluates all RNA transcripts including ncRNAs via the application of a FANTOM-CAT reference.

FANTOM-CAT is expected to provide additional insights through integration with other studies, such as its use in a comprehensive expression atlas across the broad human transcriptome called FC-R2 (Imada et al, 2020). The application of this dataset allows for a more comprehensive evaluation of changes in the gene expression data from human experiments, expanding the evaluations to the ncRNA transcripts. However, to the best of our knowledge, FANTOM-CAT is only published as a General Transfer Format (GTF) file and is not available in FASTA format; thus, the underlying data are not publicly available. As this dataset may not be commonly applied as

**Table 2. Top 25 up- and down-regulated transcripts identified using their HNf-scores.**

| | Transcript ID | HNf-score | Chr | Gene name | Gene class | Antisense |
|---|---|---|---|---|---|---|
| Top 25 up-regulated transcripts | | | | | | |
| 1 | ENST00000607600.1 | 321 | chr6 | RP1-261G23.7 | Short ncRNAs | AS_VEGFA |
| 2 | ENST00000481651.1 | 309 | chr7 | RP11-61L23.2 | Pseudogenes | |
| 3 | ENST00000307365.3 | 301 | chr10 | DDIT4 | Protein-coding mRNAs | |
| 4 | FTMT21000000269.1 | 300 | chr3 | BHLHE40-AS1 | lncRNA, intergenic | AS_BHLHE40 |
| 5 | ENST00000368636.4 | 289 | chr10 | BNIP3 | Protein-coding mRNAs | |
| 6 | ENST00000335174.4 | 275 | chr4 | ANKRD37 | Protein-coding mRNAs | |
| 7 | ENST00000290573.2 | 272 | chr2 | HK2 | Protein-coding mRNAs | |
| 8 | FTMT24300004891.1 | 267 | chr11 | LDHA | Protein-coding mRNAs | |
| 9 | HBMT00000242044.1 | 265 | chr11 | SBF2 | Protein-coding mRNAs | AS_ADM |
| 10 | FTMT24200000803.1 | 265 | chr11 | CATG00000004979.1 | lncRNA, antisense | AS_LDHA |
| 11 | ENST00000380629.2 | 265 | chr8 | BNIP3L | Protein-coding mRNAs | |
| 12 | ENCT00000004975.1 | 263 | chr1 | SLC2A1-AS1 | lncRNA, divergent | AS_SLC2A1 |
| 13 | ENST00000543445.1 | 257 | chr11 | LDHA | Protein-coding mRNAs | |
| 14 | ENST00000378357.4 | 257 | chr9 | CA9 | Protein-coding mRNAs | |
| 15 | ENST00000453116.1 | 254 | chr10 | MXI1 | Protein-coding mRNAs | |
| 16 | ENST00000471240.1 | 253 | chr10 | DDIT4 | Protein-coding mRNAs | |
| 17 | ENST00000460806.1 | 251 | chr3 | BHLHE40 | Protein-coding mRNAs | |
| 18 | FTMT20200001505.1 | 250 | chr1 | SLC2A1 | Protein-coding mRNAs | |
| 19 | HBMT00000734702.1 | 250 | chr19 | CATG00000040757.1 | lncRNA, divergent | AS_GPI |
| 20 | FTMT21100041760.1 | 249 | chr3 | BHLHE40 | Protein-coding mRNAs | |
| 21 | ENST00000250457.3 | 246 | chr14 | EGLN3 | Protein-coding mRNAs | |
| 22 | HBMT00000734730.1 | 245 | chr19 | CATG00000040757.1 | lncRNA, divergent | AS_GPI |
| 23 | ENST00000426263.3 | 243 | chr1 | SLC2A1 | Protein-coding mRNAs | |
| 24 | HBMT00000727478.1 | 242 | chr19 | RAB11B-AS1 | lncRNA, divergent | AS_ANGPTL4 |
| 25 | ENST00000534464.1 | 241 | chr11 | ADM | Protein-coding mRNAs | |
| Top 25 down-regulated transcripts | | | | | | |
| 1 | ENST00000320270.2 | −248 | chr8 | RRS1 | Protein-coding mRNAs | |
| 2 | ENST00000368232.4 | −246 | chr1 | GPATCH4 | Protein-coding mRNAs | |
| 3 | ENST00000361453.3 | −245 | chrM | MT-ND2 | Protein-coding mRNAs | |
| 4 | ENST00000361851.1 | −230 | chrM | MT-ATP8 | Protein-coding mRNAs | |
| 5 | ENST00000361227.2 | −221 | chrM | MT-ND3 | Protein-coding mRNAs | |
| 6 | ENST00000361390.2 | −215 | chrM | MT-ND1 | Protein-coding mRNAs | |
| 7 | ENCT00000264196.1 | −200 | chr20 | IDH3B | Protein-coding mRNAs | AS_NOP56 |
| 8 | ENST00000362079.2 | −194 | chrM | MT-CO3 | Protein-coding mRNAs | |
| 9 | FTMT30000000001.1 | −193 | chrM | MT-ND5 | Protein-coding mRNAs | AS_MT-ND6 |
| 10 | ENCT00000020267.1 | −188 | chr1 | CATG00000042732.1 | lncRNA, divergent | |
| 11 | FTMT20300078215.1 | −188 | chr1 | APOA1BP | Protein-coding mRNAs | AS_GPATCH4 |
| 12 | FTMT26100009680.1 | −188 | chr16 | POLR3K | Protein-coding mRNAs | |
| 13 | FTMT24100001922.1 | −187 | chr11 | H2AFX | Protein-coding mRNAs | AS_HMBS |
| 14 | MICT00000370386.1 | −185 | chrM | MT-TA | Structural RNAs | AS_MT-ATP6, etc. |
| 15 | ENST00000416718.2 | −177 | chr1 | RP5-857K21.11 | Pseudogenes | |
| 16 | HBMT00000533809.1 | −176 | chr16 | ALG1 | Protein-coding mRNAs | AS_EEF2KMT |

**Table 2.** Continued

| | Transcript ID | HNf-score | Chr | Gene name | Gene class | Antisense |
|---|---|---|---|---|---|---|
| 17 | FTMT22300045607.1 | −176 | chr6 | SRSF3 | Protein-coding mRNAs | |
| 18 | ENST00000585075.1 | −175 | chr17 | RP11-649A18.12 | lncRNA, divergent | AS_SLC25A19 |
| 19 | ENST00000295304.4 | −175 | chr2 | CHAC2 | Protein-coding mRNAs | |
| 20 | MICT00000370388.1 | −175 | chrM | MT-TA | Structural RNAs | AS_MT-ATP6, etc. |
| 21 | ENST00000361899.2 | −175 | chrM | MT-ATP6 | Protein-coding mRNAs | |
| 22 | ENST00000458605.1 | −174 | chr22 | RRP7B | Pseudogenes | |
| 23 | HBMT00000611042.1 | −172 | chr17 | METTL23 | Protein-coding mRNAs | |
| 24 | ENST00000371538.3 | −170 | chr1 | SELRC1 | Protein-coding mRNAs | |
| 25 | ENST00000293860.5 | −169 | chr16 | POLR3K | Protein-coding mRNAs | |

a reference point for RNA-seq, we investigated the number of articles referencing both "RNA-seq" and "FANTOM" in PMC.

To investigate how often RNA-seq quantification is performed using FANTOM-CAT, the number of RNA-seq publications using FANTOM-CAT and GENCODE was searched and quantified in PMC. The number of articles with both "RNA-seq" and "FANTOM" accounted for less than 1% of the total that describes "RNA-seq" data even after 2017 when the FANTOM-CAT article was first published (Table 1). These data also indicated that there were only a few reports describing gene expression quantification using FANTOM-CAT as a reference and that most studies continue to focus on more well-established gene sets, potentially overlooking critical actors in the under-evaluated data (Ono & Bono, 2021). When we combined these observations, we decided that a comprehensive analysis, including the ncRNAs, was likely worth the effort when discussing hypoxia. Given this, we went on to complete a FANTOM-CAT–based evaluation of the hypoxia data described above.

To this end, we created a new FASTA file using the FANTOM-CAT genome annotation GTF file and then added the HNf-score to each of the FANTOM-CAT lv4 transcripts (Fig 4A). Subsequent analysis of this file then revealed that the bottom 25 genes in the HNf-score dataset were predominantly mitochondrial DNA–derived transcripts. Visualization of this patterning then revealed likely regulation of gene expression by chromosome (Fig 4B and Table 2) and confirmed that these effects were most pronounced for mitochondrial transcripts. The down-regulation of mitochondrial genes such as *MT-ND2*, *MT-ATP8*, and *MT-ND* by hypoxic stimulation is already well known (Arnaiz et al, 2021), validating our data. It has also been reported (Zhang et al, 2008) that mitochondrial autophagy is commonly induced in response to hypoxia and that this process requires hypoxia-dependent factor-1–dependent BNIP3 expression, which also supports our results. On the contrary, the top 25 transcripts from the HNf-score dataset were dominated by antisense reads, such as *BHLHE40-AS1* and *SLC2A1-AS1* (Table 2). This was an interesting observation, and its implications were supported by the fact that the sense transcripts for both *BHLHE40* and *SLC2A1* are well-established hypoxia response genes. In addition, RP1-261G23.7, which had the highest HNf-score, is a short ncRNA localized to the antisense of vascular endothelial growth factor A, a well-known hypoxia response gene. This short ncRNA has been reported to be functionally involved in the transcriptional

regulation of vascular endothelial growth factor A during hypoxia (Nieminen et al, 2018). Based on these findings, we thought that the transcripts from the antisense strands of known hypoxia-responsive genes were likely worth further investigation.

Antisense transcripts are transcribed from the complementary strand of the DNA molecule and usually overlap with other sense transcripts encoding both proteins and noncoding regulators. It is also worth noting that ~30% of human (Ozsolak et al, 2010) and mouse (Katayama et al, 2005) transcripts originate from the antisense strand. Given this, we went on to select 200 hypoxia-responsive genes from both the up- and down-regulated gene sets using their HNg-score. We then evaluated the HNf-score of all antisense transcripts of these genes.

This revealed that 75% of the up- and down-regulated antisense transcripts presented with a similar pattern of expression to their sense partner, whereas the other 25% of antisense transcripts were shown to be regulated in opposition to their sense partner (Fig 5A). Several functions are regulated in the same direction. Some antisense transcripts enhance the translational function of sense transcripts or repress degradation and translational repression (Faghihi et al, 2010; Carrieri et al, 2012).

Further evaluation then revealed that the bottom 10 antisense transcripts in the DOWN200 gene set were dominated by mitochondrial genes (Table 2 and Fig 4B). Interestingly, the top 10 transcripts from the DOWN200 gene set included *PGK1* and antisense *TAF9B*, whose expression was suppressed in response to hypoxia (Fig 5C). This is of specific interest as *PGK1* catalyzes the reversible conversion of 1,3-diphosphoglycerate to 3-phosphoglycerate in the glycolytic system to generate ATP (Valentin et al, 1998), and this gene is well known for its increased expression in response to HIF1A signaling after hypoxic induction (Li et al, 1996). Meanwhile, TAF9B is one of the proteins associated with the TFIID and plays an important role in transcriptional initiation (Tora, 2002; Frontini et al, 2005). TAF9B is also known to be up-regulated in response to HIF1A knockdown and down-regulated under hypoxic conditions (Mathieu et al, 2011). Using FANTOM-CAT, the relationship between *PGK1* and *TAF9B* was shown to be that of a sense–antisense pair. This relationship is not present in either the UCSC gene or GENCODE annotations (Fig 5B).

In addition, the regulation of genes by their antisense transcripts has been extensively studied (Katayama et al, 2005; Okada et al,

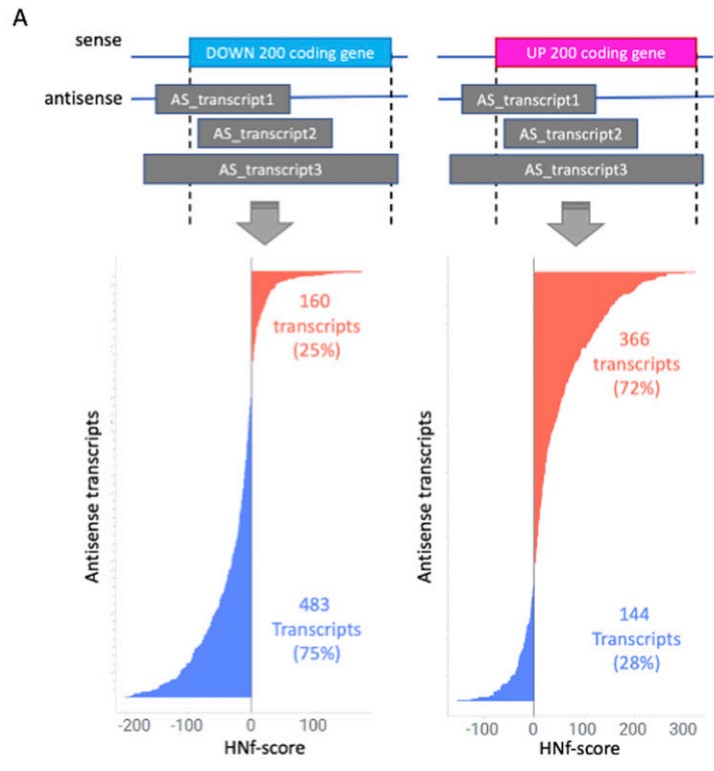

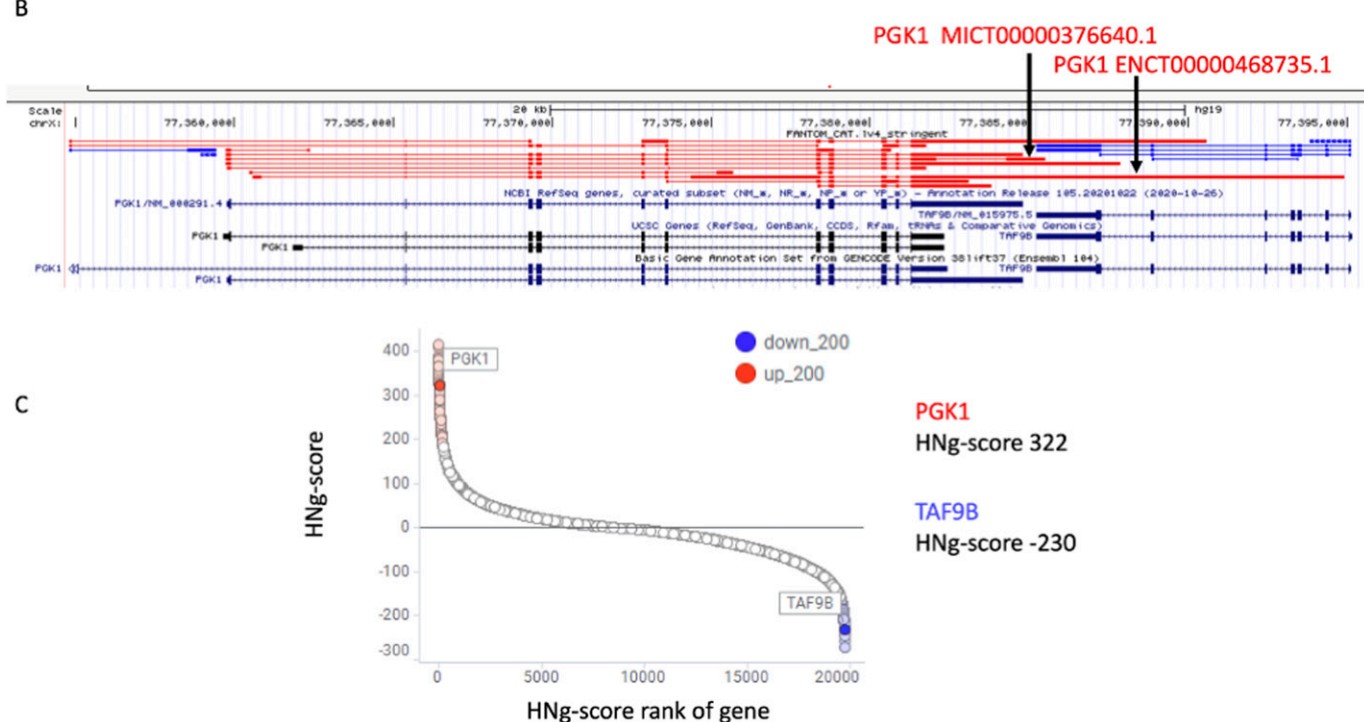

**Figure 5. Hypoxia–normoxia (HN) score of coding and noncoding genes' (HNf-score) antisense transcripts in connection to their sense counterparts.**
**(A)** Distribution of HNf-scores for each antisense transcript from each gene in the UP200 and DOWN200 gene lists. **(B)** Visualization of *PGK1* and *TAF9B* loci in the UCSC Genome Browser. In FANTOM CAGE–Associated Transcriptome, *PGK1* transcripts are a sense–antisense pair for *TAF9B*, but GENCODE does not show this relationship. **(C)** HN-score for coding genes for *PGK1* and *TAF9B*.

**Table 3.  Top and bottom 10 antisense transcripts for UP200 and DOWN200 HNg-score gene sets, as identified based on their HNf-scores.**

| | Transcript ID | HNf-score | Gene name | Gene class | Antisense |
|---|---|---|---|---|---|
| Antisense transcripts from the UP200 gene set | | | | | |
| Top 10 | | | | | |
| 1 | ENST00000607600.1 | 321 | RP1-261G23.7 | Short ncRNAs | AS_VEGFA |
| 2 | FTMT21000000269.1 | 300 | BHLHE40-AS1 | lncRNA, intergenic | AS_BHLHE40 |
| 3 | HBMT00000242044.1 | 265 | SBF2 | Protein-coding mRNAs | AS_ADM |
| 4 | FTMT24200000803.1 | 265 | CATG00000004979.1 | lncRNA, antisense | AS_LDHA |
| 5 | ENCT00000004975.1 | 263 | SLC2A1-AS1 | lncRNA, divergent | AS_SLC2A1 |
| 6 | HBMT00000734702.1 | 250 | CATG00000040757.1 | lncRNA, divergent | AS_GPI |
| 7 | HBMT00000734730.1 | 245 | CATG00000040757.1 | lncRNA, divergent | AS_GPI |
| 8 | HBMT00000727478.1 | 242 | RAB11B-AS1 | lncRNA, divergent | AS_ANGPTL4 |
| 9 | ENCT00000211059.1 | 241 | RAB11B-AS1 | lncRNA, divergent | AS_ANGPTL4 |
| 10 | HBMT00000734676.1 | 234 | CATG00000040757.1 | lncRNA, divergent | AS_GPI |
| Bottom 10 | | | | | |
| 1 | ENCT00000244163.1 | −152 | RP11-259N19.1 | lncRNA, divergent | AS_HK2 |
| 2 | HBMT00000507967.1 | −124 | CATG00000025826.1 | Protein-coding mRNAs | AS_ISG20 |
| 3 | ENST00000309424.3 | −115 | CD3EAP | Protein-coding mRNAs | AS_PPP1R13L |
| 4 | FTMT21900007493.1 | −111 | SRFBP1 | Protein-coding mRNAs | AS_LOX |
| 5 | ENST00000524270.1 | −94 | SPSB2 | Protein-coding mRNAs | AS_TPI1 |
| 6 | ENST00000357429.6 | −87 | C7orf50 | Protein-coding mRNAs | AS_GPR146 |
| 7 | ENST00000462901.1 | −81 | CGGBP1 | Protein-coding mRNAs | AS_ZNF654 |
| 8 | ENST00000439489.1 | −81 | DIABLO | Protein-coding mRNAs | AS_B3GNT4 |
| 9 | MICT00000230070.1 | −80 | PPIL2 | Protein-coding mRNAs | AS_YPEL1 |
| 10 | ENST00000422285.2 | −79 | PDHA1 | Protein-coding mRNAs | AS_MAP3K15 |
| Antisense transcripts from the DOWN200 gene set | | | | | |
| Top 10 | | | | | |
| 1 | MICT00000376640.1 | 175 | PGK1 | Protein-coding mRNAs | AS_TAF9B |
| 2 | ENCT00000468735.1 | 145 | PGK1 | Protein-coding mRNAs | AS_TAF9B |
| 3 | ENST00000229270.4 | 128 | TPI1 | Protein-coding mRNAs | AS_SPSB2 |
| 4 | FTMT29100015042.1 | 120 | PGK1 | Protein-coding mRNAs | AS_TAF9B |
| 5 | ENST00000566326.1 | 114 | MAP2K1 | Protein-coding mRNAs | AS_SNAPC5 |
| 6 | ENST00000329078.3 | 99 | SPNS2 | Protein-coding mRNAs | AS_MYBBP1A |
| 7 | ENST00000222256.4 | 87 | RAB3A | Protein-coding mRNAs | AS_MPV17L2 |
| 8 | ENCT00000059259.1 | 76 | PDZD7 | Protein-coding mRNAs | AS_TWNK |
| 9 | MICT00000139943.1 | 71 | SPNS2 | Protein-coding mRNAs | AS_MYBBP1A |
| 10 | FTMT25300024476.1 | 70 | ZBTB25 | Protein-coding mRNAs | AS_MTHFD1 |
| Bottom 10 | | | | | |
| 1 | ENCT00000264196.1 | −200 | IDH3B | Protein-coding mRNAs | AS_NOP56 |
| 2 | FTMT30000000001.1 | −193 | MT-ND5 | Protein-coding mRNAs | AS_MT-ND6 |
| 3 | FTMT20300078215.1 | −188 | APOA1BP | Protein-coding mRNAs | AS_GPATCH4 |
| 4 | FTMT24100001922.1 | −187 | H2AFX | Protein-coding mRNAs | AS_HMBS |
| 5 | MICT00000370386.1 | −185 | MT-TA | Structural RNAs | AS_MT-ATP6 |
| 6 | MICT00000370386.1 | −185 | MT-TA | Structural RNAs | AS_MT-ATP8 |
| 7 | MICT00000370386.1 | −185 | MT-TA | Structural RNAs | AS_MT-CO3 |

**Table 3. Continued**

| | Transcript ID | HNf-score | Gene name | Gene class | Antisense |
|---|---|---|---|---|---|
| 8 | MICT00000370386.1 | −185 | MT-TA | Structural RNAs | AS_MT-ND1 |
| 9 | MICT00000370386.1 | −185 | MT-TA | Structural RNAs | AS_MT-ND2 |
| 10 | MICT00000370386.1 | −185 | MT-TA | Structural RNAs | AS_MT-ND3 |

2008; Pelechano & Steinmetz, 2013), with many studies focusing on antisense transcript–mediated epigenetic regulation and the possibility that when transcription is initiated simultaneously from both sense and antisense start sites, the factors involved in their transcription may interact to repress gene expression. These results suggest that the 3′ end of *PGK1* affects the expression of *TAF9B*.

The datasets used in these evaluations were not derived from strand-specific RNA sequencing. However, these findings suggest that strand-specific RNA-seq or knockdown experiments may facilitate a better understanding of the hypoxic response and should be considered in the future.

Thus, this article describes the use of FANTOM-CAT and other open data sources to facilitate the exploratory analysis of existing data. We first evaluated the ncRNA metabolic process and provided a dataset that will help generate hypotheses for clarifying specific ncRNA intracellular responses. We used this approach to further characterize hypoxia-responsive transcripts and overcome the native bias in genome research toward known genes. Our study used the increased processing power of computers to enable the application of big data analysis to drive discovery. We believe that this data-driven approach is increasingly necessary to help advance genomic research. Our data suggest that FANTOM-CAT has not been widely used in research despite its obvious value. Thus, we designed this study to showcase the value of FANTOM-CAT–derived transcript annotation as a means to uncover new targets for investigation and validation. Our evaluation of the hypoxia data uncovered a novel connection between *PGK1* and *TAF9B* which was not reported in any other evaluations despite their critical value in hypoxia.

Thus, we suggest that this approach can be applied to other questions to facilitate more nuanced and less biased evaluations of large RNA-seq datasets and propose that FANTOM-CAT should be commonly evaluated as an indispensable tool in transcriptome evaluations.

# Materials and Methods

### Evaluating the number of articles with descriptions of FANTOM

We used the following queries across our PMC web search (conducted on 9 April 2022):

RNA-Seq: "rna-seq"[MeSH Terms] OR "rna-seq"[All Fields] OR ("rna"[All Fields] AND "seq"[All Fields]) OR "rna seq"[All Fields]. RNA-Seq AND GENCODE: ("rna-seq"[MeSH Terms] OR "rna-seq"[All Fields] OR ("rna"[All Fields] AND "seq"[All Fields]) OR "rna seq"[All Fields]) AND "gencode"[All Fields].

RNA-Seq AND FANTOM: ("rna-seq"[MeSH Terms] OR "rna-seq"[All Fields] OR ("rna"[All Fields] AND "seq"[All Fields]) OR "rna seq"[All Fields]) AND "fantom"[All Fields] to identify any relevant publications. Once identified, we downloaded the relevant article information in MEDLINE format and calculated the number of publications per year based on the date of publication.

### Public gene expression data

We then collected the RNA-seq data (https://doi.org/10.6084/m9.figshare.14141219.v1) from 495 pairs of human hypoxia–normoxia samples evaluated in our previous study (Ono & Bono, 2021).

### Gene expression quantification

The expression of coding genes was quantified using ikra 1.2.3 (https://github.com/yyoshiaki/ikra) as previously reported. The following procedure was used to quantify gene expression when FANTOM-CAT was used as the reference sequence. As there were no FASTA-formatted files for these data recorded in the FANTOM-CAT repository, we created a FASTA file based on the GTF file data using gffread v0.12.1 and the following command:

%gffread FANTOM_CAT.lv4_stringent.only_lncRNA.gtf -g hg19.fa -w lv4.fa.

We used salmon 0.14.0 for index creation and quantification (https://github.com/no85j/hypoxia_code/tree/master/salmon/salmon_v0.14.0), with some modifications to Pitagora-cwl (https://github.com/pitagora-network/pitagora-cwl/tree/master/tools/salmon) and tximport 1.18.0 (https://github.com/no85j/hypoxia_code/tree/master/tximport) to match as closely as possible with the quantification method used in our previous studies (Bono & Hirota, 2020; Ono & Bono, 2021). We then calculated scaledTPM for use this as a representative value for gene expression (Soneson et al, 2016). Our quantitative RNA-seq data are accessible at figshare (https://doi.org/10.6084/m9.figshare.19679520.v1).

### HN-score

The fold change of the [scaledTPM] + 1 values for each of the 495 sample pairs was calculated, and differential expression was determined at a 1.5-fold cutoff. We then produced the hypoxia–normoxia (HN) score for this data by taking the number of samples demonstrating significant up-regulation and subtracting the number of samples with significant down-regulation of their overall expression profile. Here, we added a second HN-score calculation method using different reference transcripts to allow for a more nuanced evaluation. These scores were then referred to as HNg (coding genes) and HNf (coding and noncoding gene), respectively. This first score was designed to evaluate the coding gene and was calculated from the previously published dataset (https://doi.org/10.6084/m9.figshare.14141135.v1) and the previously described

HN1.5 cutoff value. This means that the HNf-score is effectively a descriptor of the HN-score for any FANTOM-CAT transcripts. This HNf-score was calculated as described above except that FANTOM-CAT was used as a reference. Transcript annotation was obtained from the FANTOM-CAT website (https://fantom.gsc.riken.jp/cat/v1/#/genes), and the HNf-score data are accessible at figshare (https://doi.org/10.6084/m9.figshare.19679508.v2).

### Enrichment analysis

We created UP200 and Down200 gene sets for the top and bottom 200 gene lists of the HNg-score, respectively, and performed enrichment analysis in Metascape (https://metascape.org/) (Zhou et al, 2019). We used the default parameters for Metascape settings.

### Antisense transcript

We used the bedtools (v2.30.0) program to complete the following evaluations (Quinlan & Hall, 2010), including the creation of a bed file in which all the regions of each gene were merged based on the transcript id of the UP200 and DOWN200 genes identified in the coding gene dataset. We then used this bed file to identify the transcripts in the reverse strand using the intersect command.

### Visualization

We used TIBCO Spotfire Desktop version 11.5.0 (TIBCO Spotfire, Inc.) to produce all of our bar and scatter plots, and we used the UpSetR (1.4.0) package from R (4.0.3) to create our UpSet plots. We then used the pandas (1.2.5), matplotlib (3.2.2), and seaborn (0.11.0) packages in Python (3.8.10) to complete our violin plotting and visualized the genome coordinates by adding "FANTOM5 summary Tracks" to the UCSC Genome Browser.

## Data Availability

Identifiers of the RNA-seq data used in this study are listed in the link to the left (https://doi.org/10.6084/m9.figshare.14141219.v1).

## Supplementary Information

## Acknowledgements

The authors would like to thank Drs. Takeya Kasukawa and Masaki Suimye Morioka for their input and discussion.

### Author Contributions

Y Ono: conceptualization, resources, data curation, software, formal analysis, validation, investigation, visualization, methodology, and writing—original draft.

H Bono: conceptualization, resources, formal analysis, supervision, funding acquisition, validation, project administration, and writing—review and editing.

### Conflict of Interest Statement

The authors declare that they have no conflict of interest.

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
