## [Reviewer comments · Life Science Alliance]

Life Science Alliance

Exploratory Meta-Analysis of Hypoxic Transcriptomes Using Precise Transcript Reference Sequence Set

Yoko Ono and Hidemasa Bono

DOI: 10.26508/lsa.202201518

Corresponding author(s): Hidemasa Bono, Hiroshima University and Yoko Ono, Hiroshima University

Review Timeline:

Submission Date:	2022-05-09
Editorial Decision:	2022-06-23
Revision Received:	2022-09-08
Editorial Decision:	2022-09-23
Revision Received:	2022-09-23
Accepted:	2022-09-26

Scientific Editor: Novella Guidi

Transaction Report:

June 23, 2022

Re: Life Science Alliance manuscript #LSA-2022-01518-T

Hidemasa Bono
Hiroshima University

Dear Dr. Bono,

Thank you for submitting your manuscript entitled "Exploratory Meta-Analysis of Hypoxic Transcriptomes Using a Precise Transcript Reference Sequence Set" to Life Science Alliance. The manuscript was assessed by expert reviewers, whose comments are appended to this letter. We invite you to submit a revised manuscript addressing the Reviewer comments.

Thank you for this interesting contribution to Life Science Alliance. We are looking forward to receiving your revised manuscript.

Sincerely,

B. MANUSCRIPT ORGANIZATION AND FORMATTING:

Reviewer #1 (Comments to the Authors (Required)):

The manuscript by Ono and Bono describes a novel approach to exploring hypoxia-responsive genes by meta-analysis. Using two reference datasets such as GENCODE and FANTOM-CAT, top and bottom genes were identified based on HNg-score, and enrichment analysis showed enriched GO in top and bottom genes. Using FANTOM-CAT data, HNg-score was evaluated, and the relationship between antisense and sense genes in down- and up-regulated genes by hypoxia. The approach is interesting because this could be robust and novel; however, there are several concerns before considering further process. Specific comments are shown below.

1. In this study, hypoxia-responsive genes were identified by meta-analysis. However, the hypoxia-responsive genes might be changed in a tissue-dependent manner. Would it be possible to show the hypoxia-related genes based on tissue types or cell types?
2. It is unclear the reason to show "Survey of the number of papers referring FANTOM".
3. As shown in Figure 4, some sense-antisense pairs expressed oppositely. Is there any explanation for this phenotype? Have the authors examined chromatin state by ATAC-seq or ChIP-seq analysis? It would be more informative to show chromatin status based on the classified four categories.
4. It may be switched the position of Figure 2A and 2B because Figure 2B was explained first in the text.
5. Figure 2C and 2D did not appear in the text.

Reviewer #2 (Comments to the Authors (Required)):

In this report, Ono et al investigated Hypoxia RNA-seq datasets and evaluated the effects on non-coding RNAs using integration with Fantom capabilities. They identify a significant level of hypoxia on the expression of non-coding RNAs, mostly the antisense variety. In addition, they identified how their regulations compares to their sense version. This study adds new insights into the mechanisms controlled by oxygen reduction in the cell. The data is robust and significant. However, my only concern is the lack of discussion on how these non-coding RNAs are influencing cellular response. In the anti-sense RNAs that share same control as their sense conter-part is there a known function? RNA-decay, chromatin organisation, and specific sequence or motif that is over represented? Also, I assume these are shared amongst all the RNA-seq experiment analysed or are they only present in a subset? If so, is there a tissue specificity or cell background specificity?

Reviewer #3 (Comments to the Authors (Required)):

The manuscript entitled "Exploratory Meta-Analysis of Hypoxic Transcriptomes Using a Precise Transcript Reference Sequence Set" by Ono et al, describes the characterization of the transcriptional response to hypoxia both at the level of coding and non-coding genes. To this end, the authors perform two analyses, one using RNAseq data quantified using the GENCODE transcriptome reference, and then a second one using the expanded meta-transcriptome assembly produce by the FANTOM consortium (the FANTOM-CAT reference), which accounts for a large collection of genes, encompassing both coding and non-coding, including many that are novel. In this study, the authors build upon their previous meta-analysis of the expression response to hypoxia, which revealed that many genes related to processing non-coding RNA were among those modulated by hypoxia. First the authors confirm their previous findings using coding genes from GENCODE, identifying pathways and processes downstream hypoxia, then they repeat the analysis with the FANTOM-CAT reference, quantifying the ncRNA associated with hypoxia, many of which are anti-sense transcripts of known coding genes.

This work is mostly descriptive and there are several aspects that need improvement, especially in terms of quantitative versus qualitative analyses, as explained below. Furthermore, the use of the FANTOM-CAT transcriptome to quantify gene expression using data from the public domain is not novel (see PMID:32079618, "Recounting the FANTOM CAGE-Associated Transcriptome" by Imada et al). Such FANTOM-CAT-recount2 gene expression atlas has been already successfully used, for instance, to quantify gene expression in prostate cancer (PMID:34311724, "Transcriptional landscape of PTEN loss in primary prostate cancer" by Imada et al), and to assist assigning a potential function to novel lncRNAs by the FANTOM6 consortium (PMID:32718982, "Functional annotation of human long noncoding RNAs via molecular phenotyping", by Ramilowski et al). The authors should include these citations and acknowledge previous work in the field.

Below is a summary of some of the limitations:

Page 6, lines 101-103, Content: "These evaluations revealed a decrease in mitochondrial DNA-derived transcripts while antisense analysis suggests a clear relationship between transcriptional regulation for both the sense and antisense genes (PGK1 and TAF9B)."

Comment: Is this relationship global or just for these 2 genes? This sentence is not really clear, please explain

Page 6, lines 106-107, Content: "Enrichment analysis focusing only on protein-coding genes revealed that the top 200 genes identified by HNg-score, a measure of susceptibility to hypoxia, were enriched for hypoxia response-"

Comment: This is to be expected. Can the authors i elaborate on the way this HN score is calculated? Also, is the correct abbreviation for the score HN, HNg, or HNf (see fig 1)

Page 7, lines 111-113, Content: "The ncRNA metabolic process was then further subdivided into the following gene ontology terms: ncRNA processing, ribosome biogenesis, ribonucleoprotein complex biogenesis, rRNA processing, and rRNA metabolic process."

Comment: It look like this sub-division is based on gene ontology however it is not clear if the GO terms indicated here resulted from some formal analysis, or whether they were selected with on arbitrary approach. Please explain.

Page 7, line 118, Content: "(A, B) Enrichment analysis for (A) the 200 DOWN regulated gene list and (B) the 200 UP regulated gene list. (C, D) GO network (C) "

Comment: What is the rationale to focus only on the first 200 genes here? Is this an arbitrary choice? What enrichment test did the authors used? Wouldn't GSEA (based on the ranking of all genes) be more appropriate? For this latter approach, no arbitrary choices are required to select the top most up- or down-regulated genes.

Page 7, line 119, Content: "Upset plot (D)"

Comment: In figure 2 panel D, it is not clear what the histograms represents? Count of genes in the intersections? Gene sets? Please explain and clarify.

Page 8, line 123, Content: "Survey of the number of papers referring FANTOM"

Comment: It is not sure what this survey is telling the reader, why it is important here? What does this information tells us? Please make an effort to put this survey into context, also it does not appear to include recent paper in which the FANTOM-CAT meta-transcriptome was already used for gene expression analysis (see PMID:32079618, PMID:34311724, and PMID:32718982, for instance). The authors should include these citations and acknowledge previous work.

Page 9, line 132, Content: "The HNf-score for each transcript was quantified using FANTOM-CAT as a reference."

Comment: Was this done for all transcripts and genes in this reference transcriptome, or only for the non-coding ones?

Page 9, lines 137-139, Content: "This investigation revealed the fact that there were several cases where hypoxia-responsive transcripts with high HNf-scores were located within the antisense regions of genes with high HNg- scores (BHLHE40-AS1, SLC2A1-AS1)."

Comment: Does this mean that both sense and anti-sense genes are affected by hypoxia? If so, is this observed more often than one would expect by chance? Is this observation only qualitative? Can the authors perform a more quantitative analysis on this point?

Page 12, lines 159-161, Content: "Further evaluation also revealed that the Top 10 antisense transcripts in the UP200 gene list were dominated by both short and long ncRNAs (lncRNAs), while the bottom 10 transcripts from this group were shown to be predominantly mRNA transcripts."

Comment: Is this different from what one would be expecting by chance alone? In other words, is there a way to test if this is relevant and not just anecdotal? See my previous comment about this above.

Page 12, lines 162-170, Content: "Interestingly, among the antisense transcripts from the DOWN200 gene set, phosphoglycerate kinase 1 (PGK1), which was also included in the UP200 gene set, had the highest HNf-score. Given this, we homed in on PGK1 and its antisense TATA-box binding protein associated factor 9b (TAF9B), evaluating their chromosomal localization using UCSC Genome Browser and HN-score at the gene level (Fig. 4B, C). This analysis revealed that while these two genes are not described as having a sense-antisense relationship when evaluated using the UCSC or GENCODE datasets, FANTOM-CAT-based analysis suggests that these two transcripts are actually in a sense-antisense relationship. By using FANTOM-CAT as a reference, we were able to present the possibility that these two genes affect the regulation of expression."

Comment: It is not clear the significance of this finding. Why is this important? Isn't this to be expected given that the ANTOM-CAT meta-transcriptome contains more genes, both coding and non-coding?

Pages 16-17, lines 195-197, Content: "and then we expanded this evaluation to any of the hypoxia-responsive genes annotated in the FANTOM-CAT database, which includes both coding and non-coding elements (Fig. 1)"

Comment: It is not clear why the author decided to repeat the analysis twice, and not just once with the FANTOM-CAT reference, which contains both coding and non-coding genes, and encompasses GENCODE. A better approach would be to perform the analysis just once, with the FANTOM-CAT, splitting coding and non-coding genes.

Page 17, lines 198-199, Content: "We believe that these data sets and this analysis approach will help to further elucidate the mechanisms regulating the hypoxic response."

Comment: Would it be possible to go a little further here and try to annotate these genes looking at other conditions for instance tissue expression in gtex, on other data sets... The FANTOM-CAT-recount2 gene expression atlas would be ideal for this analysis.

Page 18, lines 224-225, Content: "Therefore, we assumed that this dataset is unlikely to be commonly applied as a reference point for RNA-seq."

Comment: This is not true, it has been used already multiple times, see my comment above.

Page 20, lines 262-264, Content: "This revealed that 75% of the UP and DOWN regulated antisense transcripts presented with a similar pattern of expression to their sense partner, while the other 25% of antisense transcripts were shown to be regulated in opposition to their sense partner (Fig. 4A)."

Comment: Is this true only for the hypoxia genes or for all genes? This statement should be supported by a formal quantitative analysis.

Finally, code and scripts used for preprocessing and analyze the gene expression data should be included in the interest of robustness, transparency, and reproducibility. The authors could use a gitHub repository or include the scripts as supplementary material.

Here are some additional minor comments:

Page 3, line 45, Content: "requires a move away"

Comment: Please rephrase: "moving away" or "to move away"

Page 5, line 76, Content: "overarching results for each chromosome."

Comment: Not sure what "overarching" means here

Page 5, line 82, Content: "Our data describe various analyses"

Comment: Not sure what this means, please rephrase

Page 5, line 93, Content: "data set"

Comment: Use "dataset"

Page 9, lines 145-146, Content: "(A) Procedure for calculating HNF-score using FANTOM-CAT as a reference. "

Comment: I am not sure that presenting the alignment procedure here and in this form is useful. Panel A does not provide much useful information, this information should be in the methods.

Reviewer #1 (Comments to the Authors (Required)):

The manuscript by Ono and Bono describes a novel approach to exploring hypoxia-responsive genes by meta-analysis. Using two reference datasets such as GENCODE and FANTOM-CAT, top and bottom genes were identified based on HNg-score, and enrichment analysis showed enriched GO in top and bottom genes. Using FANTOM-CAT data, HNg-score was evaluated, and the relationship between antisense and sense genes in down- and up-regulated genes by hypoxia. The approach is interesting because this could be robust and novel; however, there are several concerns before considering further process. Specific comments are shown below.

1. In this study, hypoxia-responsive genes were identified by meta-analysis. However, the hypoxia-responsive genes might be changed in a tissue-dependent manner. Would it be possible to show the hypoxia-related genes based on tissue types or cell types?

Response1-1: After Fig. 1 the sample summary data are described in detail, and gene expression information by tissue under representative conditions is also provided. (Added as Fig 2)

" The hypoxic conditions of the analyzed data were summarized for 495 samples (Fig. 2A, B). Normoxic conditions were considered as a 20% oxygen concentration, although some samples were not mentioned, whereas hypoxic conditions ranged from 0.1 to 5% oxygen concentrations and some cases of hypoxia induced by chemicals such as CoCl₂. The treatment time ranged from 1 h to 3 months. The most common condition of hypoxia in the dataset was a 1% oxygen concentration for 24 h of treatment. Sixty-five percent of all samples were of cancer origin, with breast cancer as the most common tissue of origin. Gene expression in each tissue under representative conditions (cancer, oxygen concentration 1%, 24 h treatment) is shown as the log₂-transformed fold-change compared to under normoxic condition (Fig. 2C). " (Lines 91-100)

2. It is unclear the reason to show "Survey of the number of papers referring FANTOM".

Response1-2: We added following sentence." To investigate how often RNA-seq quantification is performed using FANTOM-CAT, the number of RNA-seq publications

using FANTOM-CAT and GENCODE was searched and quantified in PMC." (Lines 239-240)

3. As shown in Figure 4, some sense-antisense pairs expressed oppositely. Is there any explanation for this phenotype? Have the authors examined chromatin state by ATAC-seq or ChIP-seq analysis? It would be more informative to show chromatin status based on the classified four categories.

Response1-3: A reference to epigenetic control has been added as follows.

"In addition, the regulation of genes by their antisense transcripts has been extensively studied (Pelechano and Steinmetz 2013; Okada et al. 2008; Katayama et al. 2005), with many studies focusing on antisense transcript-mediated epigenetic regulation and the possibility that when transcription is initiated simultaneously from both sense and antisense start sites, the factors involved in their transcription may interact to repress gene expression. These results suggest that the 3' end of *PGK1* affects the expression of *TAF9B*." (Lines 307-312)

We are convinced that epigenetic analysis will provide us with a great deal of new knowledge. However, ATAC-seq and ChIP-seq data under hypoxic treatment conditions are not included in this analysis and have not been analyzed this time.

4. It may be switched the position of Figure 2A and 2B because Figure 2B was explained first in the text.

Response1-4: Thank you for pointing this out. We have switched the position of the figures.

5. Figure 2C and 2D did not appear in the text.

Response1-5: Thank you for pointing this out. We have included these (now Figure 3C and 3D) in the text of the relevant section. "In this enrichment analysis, the ncRNA metabolic process was further subdivided into the following Gene Ontology (GO) terms: ncRNA processing, ribosome biogenesis, ribonucleoprotein complex biogenesis, rRNA processing, and rRNA metabolic process (Fig. 3C). The results of rRNA GO network analysis were visualized in Upset plots, which showed that the ncRNA metabolic process is heavily reliant on most of these rRNA metabolic genes (Fig. 3D)." (Lines 121-125)

Reviewer #2 (Comments to the Authors (Required)):

In this report, Ono et al investigated Hypoxia RNA-seq datasets and evaluated the effects on non-coding RNAs using integration with Fantom capabilities. They identify a significant level of hypoxia on the expression of non-coding RNAs, mostly the antisense variety. In addition, they identified how their regulations compares to their sense version. This study adds new insights into the mechanisms controlled by oxygen reduction in the cell. The data is robust and significant. However, my only concern is the lack of discussion on how these non-coding RNAs are influencing cellular response.

Response2-1: Thank you for your suggestion. We could not clarify the intracellular response of non-coding RNAs (ncRNAs), but it is worthwhile to evaluate the hypoxic response of transcripts including ncRNAs comprehensively by using FANTOM-CAT as a reference sequence. Based on your suggestion, we have added the following. "We first evaluated the ncRNA metabolic process and provided a dataset that will help generate hypotheses for clarifying specific ncRNA intracellular responses." (Lines 319-320)

In the anti-sense RNAs that share same control as their sense conter-part is there a known function? RNA-decay, chromatin organisation, and specific sequence or motif that is over represented?

Response2-2: Several functions are known to be regulated in the same direction. We added following sentence. "Several functions are regulated in the same direction. Some antisense transcripts enhance the translational function of sense transcripts or repress degradation and translational repression (Carrieri et al. 2012; Faghihi et al. 2010)." (Lines 287-289)

Also, I assume these are shared amongst all the RNA-seq experiment analysed or are they only present in a subset? If so, is there a tissue specificity or cell background specificity?

Response2-3: Fig 2C was newly appended to visualize gene expression for each of the PGK1 conditions. As you can see, there was no specificity in a specific condition. We believe it depends on oxygen concentration, treatment time, and cell type.

Reviewer #3 (Comments to the Authors (Required)):

The manuscript entitled "Exploratory Meta-Analysis of Hypoxic Transcriptomes Using a

Precise Transcript Reference Sequence Set" by Ono et al, describes the characterization of the transcriptional response to hypoxia both at the level of coding and non-coding genes. To this end, the authors perform two analyses, one using RNAseq data quantified using the GENCODE transcriptome reference, and then a second one using the expanded meta-transcriptome assembly produce by the FANTOM consortium (the FANTOM-CAT reference), which accounts for a large collection of genes, encompassing both coding and non-coding, including many that are novel. In this study, the authors build upon their previous meta-analysis of the expression response to hypoxia, which revealed that many genes related to processing non-coding RNA were among those modulated by hypoxia. First the authors confirm their previous findings using coding genes from GENCODE, identifying pathways and processes downstream hypoxia, then they repeat the analysis with the FANTOM-CAT reference, quantifying the ncRNA associated with hypoxia, many of which are anti-sense transcripts of known coding genes.

This work is mostly descriptive and there are several aspects that need improvement, especially in terms of quantitative versus qualitative analyses, as explained below. Furthermore, the use of the FANTOM-CAT transcriptome to quantify gene expression using data from the public domain is not novel (see PMID:32079618, "Recounting the FANTOM CAGE-Associated Transcriptome" by Imada et al). Such FANTOM-CAT-recount2 gene expression atlas has been already successfully used, for instance, to quantify gene expression in prostate cancer (PMID:34311724, "Transcriptional landscape of PTEN loss in primary prostate cancer" by Imada et al), and to assist assigning a potential function to novel lncRNAs by the FANTOM6 consortium (PMID:32718982, "Functional annotation of human long noncoding RNAs via molecular phenotyping", by Ramilowski et al). The authors should include these citations and acknowledge previous work in the field.

Thank you for your suggestions. We addressed all your comments below.

Below is a summary of some of the limitations:

Page 6, lines 101-103, Content: "These evaluations revealed a decrease in mitochondrial DNA-derived transcripts while antisense analysis suggests a clear relationship between transcriptional regulation for both the sense and antisense genes (PGK1 and TAF9B)."

Comment: Is this relationship global or just for these 2 genes? This sentence is not really clear, please explain

Response 3-1: Mitochondrial DNA's are global, and the description about PGK1 and TAF9B refers to the two genes only. The sentence correspondence was inappropriate, so the wording has been changed. "Number of transcripts derived from mitochondrial DNA and antisense analysis was reduced, suggesting a transcriptional regulation relationship in both sense and antisense genes (*PGK1* and *TAF9B*). "(Lines 524-526)

Page 6, lines 106-107, Content: "Enrichment analysis focusing only on protein-coding genes revealed that the top 200 genes identified by HNg-score, a measure of susceptibility to hypoxia, were enriched for hypoxia response-"

Comment: This is to be expected. Can the authors i elaborate on the way this HN score is calculated? Also, is the correct abbreviation for the score HN, HNg,or Hnf (see fig 1)

Response3-2: The following text has been added for the clarification.

"From a sample of hypoxia-related datasets obtained from public databases, we selected sample pairs of hypoxia and normoxia (HN-pairs). Based on these HN-pairs, plus 1 was calculated for each HN-pair if the expression variation was greater than 1.5-fold compared to both and minus 1 if the variation was less than the reciprocal of 1.5-fold. These values were summed for each gene and designated as the hypoxia-normoxia score (HN-score). The HN-score qualitatively reflects hypoxia-responsiveness, unlike the method used in general gene expression analysis. This HN-score was named as the HNg-score when the reference sequence was GENCODE and Hnf-score when the sequence was based on FANTOM-CAT. We subjected 200 genes with high HNg-scores and 200 genes with low HNg-scores to enrichment analysis and evaluated whether the genes selected based on the HNg-score could be used to examine hypoxia responsiveness."(Lines 102-111)

Page 7, lines 111-113, Content: "The ncRNA metabolic process was then further subdivided into the following gene ontology terms: ncRNA processing, ribosome biogenesis, ribonucleoprotein complex biogenesis, rRNA processing, and rRNA metabolic process."

Comment: It look like this sub-division is based on gene ontology however it is not clear if the GO terms indicated here resulted from some formal analysis, or whether they were selected with on arbitrary approach. Please explain.

Response3-3: It is presented based on the results obtained from this analysis. The following explanation has been added. "In enrichment analysis, the ncRNA metabolic process was further subdivided into..."(Lines 121-125)

Page 7, line 118, Content: "(A, B) Enrichment analysis for (A) the 200 DOWN regulated gene list and (B) the 200 UP regulated gene list. (C, D) GO network (C) "

Comment: What is the rationale to focus only on the first 200 genes here? Is this an arbitrary choice? What enrichment test did the authors used? Wouldn't GSEA (based on the ranking of all genes) be more appropriate? For this latter approach, no arbitrary choices are required to select the top most up- or down-regulated genes.

Response3-4: Gene Set Enrichment Analysis (GSEA) is excellent method in some analyses because it can be evaluated based on gene expression levels without arbitrariness. However, it is not suitable in this case because the data used in this analysis was not based on gene expression levels commonly used in GSEA, but on HNg-scores of approximately 500 hypoxia-normal oxygen pairs. Therefore, we considered it was appropriate to select 200 genes for enrichment analysis, although arbitrariness remains. The selected 200 genes are shown in red or blue in Fig. 5C. These genes are considered to be those that are more likely to fluctuate under hypoxic conditions.

Page 7, line 119, Content: "Upset plot (D)"

Comment: In figure 2 panel D, it is not clear what the histograms represents? Count of genes in the intersections? Gene sets? Please explain and clarify.

Response3-5: We changed the description to "Upset plot showing the number of genes included in GO in this analysis" (Lines 534-535)

Page 8, line 123, Content: "Survey of the number of papers referring FANTOM"

Comment: It is not sure what this survey is telling the reader, why it is important here? What does this information tells us? Please make an effort to put this survey into context, also it does not appear to include recent paper in which the FANTOM-CAT meta-transcriptome was already used for gene expression analysis (see PMID:32079618, PMID:34311724, and PMID:32718982, for instance). The authors should include these citations and acknowledge previous work.

Response3-6: Thank you for pointing this out. We added the following sentence to ensure that it was appropriately worded.

"FANTOM-CAT is a comprehensive catalog of human lncRNAs that improves upon existing lncRNA transcript models. The results obtained using this dataset have greatly contributed to life science research (Imada et al. 2020; Ramilowski et al. 2020; Imada et al.

2021). Despite the usefulness of this dataset, we predicted that it is used less frequently for RNA-sequencing (RNA-seq) quantification compared to GENCODE; we quantitatively examined this hypothesis. If FANTOM-CAT is used less frequently, results obtained using this database may show a lower influence of publication bias (Ono and Bono 2021), as demonstrated in our previous report." (Lines 127-133)

Page 9, line 132, Content: "The HNF-score for each transcript was quantified using FANTOM-CAT as a reference."

Comment: Was this done for all transcripts and genes in this reference transcriptome, or only for the non-coding ones?

Response3-7: We quantified for all transcripts. The text was changed as follows.

"HNF-scores of all transcripts, including ncRNAs, were quantified using FANTOM-CAT as a reference." (Lines 146-147)

Page 9, lines 137-139, Content: "This investigation revealed the fact that there were several cases where hypoxia-responsive transcripts with high HNF-scores were located within the antisense regions of genes with high HNG- scores (BHLHE40-AS1, SLC2A1-AS1)."

Comment: Does this mean that both sense and anti-sense genes are affected by hypoxia? If so, is this observed more often than one would expect by chance? Is this observation only qualitative? Can the authors perform a more quantitative analysis on this point?

Response3-8: As the reviewer pointed out, this could be a coincidental result. In this case, we confirmed the antisense one by one in an exhaustive manner. We added following sentence. "We investigated each antisense transcript individually, as we could not perform enrichment analysis which requires information comprehensively evaluated the effects of sense and antisense transcripts." (Lines 159-160)

Page 12, lines 159-161, Content: "Further evaluation also revealed that the Top 10 antisense transcripts in the UP200 gene list were dominated by both short and long ncRNAs (lncRNAs), while the bottom 10 transcripts from this group were shown to be predominantly mRNA transcripts."

Comment: Is this different from what one would be expecting by chance alone? In other words, is there a way to test if this is relevant and not just anecdotal? See my previous comment about this above.

Response3-9: Since the HNF-score was used as the index for evaluation, the results are not coincidental; the following was added to indicate that the analysis was based on the HNF-score. "We also investigated transcripts with high or low HNF-scores to qualitatively evaluate hypoxia responsiveness.." (Lines 176-177)

Page 12, lines 162-170, Content: "Interestingly, among the antisense transcripts from the DOWN200 gene set, phosphoglycerate kinase 1 (PGK1), which was also included in the UP200 gene set, had the highest HNF-score. Given this, we homed in on PGK1 and its antisense TATA-box binding protein associated factor 9b (TAF9B), evaluating their chromosomal localization using UCSC Genome Browser and HN-score at the gene level (Fig. 4B, C). This analysis revealed that while these two genes are not described as having a sense-antisense relationship when evaluated using the UCSC or GENCODE datasets, FANTOM-CAT-based analysis suggests that these two transcripts are actually in a sense-antisense relationship. By using FANTOM-CAT as a reference, we were able to present the possibility that these two genes affect the regulation of expression."

Comment: It is not clear the significance of this finding. Why is this important? Isn't this to be expected given that the FANTOM-CAT meta-transcriptome contains more genes, both coding and non-coding?

Response3-10: An interesting aspect of this analysis is that relationships not found by GENCODE or UCSC were only be revealed by utilizing FANTOM-CAT. We have changed the wording as follows.

"By using FANTOM-CAT as a reference, we were able to present for the first time the possibility that these two genes affect the regulation of expression." (Lines 187-188)

Pages 16-17, lines 195-197, Content: "and then we expanded this evaluation to any of the hypoxia-responsive genes annotated in the FANTOM-CAT database, which includes both coding and non-coding elements (Fig. 1)"

Comment: It is not clear why the author decided to repeat the analysis twice, and not just once with the FANTOM-CAT reference, which contains both coding and non-coding genes, and encompasses GENCODE. A better approach would be to perform the analysis just once, with the FANTOM-CAT, splitting coding and non-coding genes.

Response3-11: As you pointed out, it would be ideal to perform the analysis using FANTOM-CAT from the beginning. However, we had already conducted our analysis focusing only on coding genes in our previous studies. Therefore, we decided to analyze

FANTOM-CAT after carefully examining the results of GENCODE first. In order to accurately describe the results, we have described them as follows.

"Our previous studies focusing on coding genes have revealed a clear publication bias when evaluating gene expression changes in response to hypoxia and identified a novel set of largely uncharacterized hypoxia-responsive elements (Ono and Bono 2021)
"(Lines 195-197)

Page 17, lines 198-199, Content: "We believe that these data sets and this analysis approach will help to further elucidate the mechanisms regulating the hypoxic response."

Comment: Would it be possible to go a little further here and try to annotate these genes looking at other conditions for instance tissue expression in gtex, on other data sets... The FANTOM-CAT-recount2 gene expression atlas would be ideal for this analysis.

Response3-12: We cited FANTOM-CAT-recount2 paper as follows. "FANTOM-CAT is expected to provide additional insights through integration with other studies, such as its use in a comprehensive expression atlas across the broad human transcriptome called FC-R2 (Imada et al. 2020). (Lines 227-229)

Page 18, lines 224-225, Content: "Therefore, we assumed that this dataset is unlikely to be commonly applied as a reference point for RNA-seq."

Comment: This is not true, it has been used already multiple times, see my comment above.

Response3-13: The wording was changed as follows.

" As this dataset may not be commonly applied as a reference point for RNA-seq, we investigated the number of articles referencing both "RNA-seq" and "FANTOM" in PMC."(Lines 235-236)

Page 20, lines 262-264, Content: "This revealed that 75% of the UP and DOWN regulated antisense transcripts presented with a similar pattern of expression to their sense partner, while the other 25% of antisense transcripts were shown to be regulated in opposition to their sense partner (Fig. 4A)."

Comment: Is this true only for the hypoxia genes or for all genes? This statement should be supported by a formal quantitative analysis.

Response3-14: The analysis is for all transcripts. The wording has been changed as follows.

"We then evaluated the HNF-score of all antisense transcripts of these genes."(Lines

283-284)

Finally, code and scripts used for preprocessing and analyze the gene expression data should be included in the interest of robustness, transparency, and reproducibility. The authors could use a gitHub repository or include the scripts as supplementary material.

Response3-15: Thanks for pointing this out. We have included the GitHub URL.

"The expression of coding genes was quantified using ikra 1.2.3 (<https://github.com/yyoshiaki/ikra>) as previously reported. The following procedure was used to quantify gene expression when FANTOM-CAT was used as the reference sequence. As there were no FASTA-formatted files for these data recorded in the FANTOM-CAT repository, we created a FASTA file based on the General Transfer Format file data using gffread v0.12.1 and the following command:

```
> gffread FANTOM_CAT.lv4_stringent.only_lncRNA.gtf -g hg19.fa -w lv4.fa
```

We used salmon 0.14.0 for index creation and quantification (https://github.com/no85j/hypoxia_code/tree/master/salmon/salmon_v0.14.0), with some modifications to Pitagora-cwl (<https://github.com/pitagora-network/pitagora-cwl/tree/master/tools/salmon>), and tximport 1.18.0 (https://github.com/no85j/hypoxia_code/tree/master/tximport) to match as closely as possible with the quantification method used in our previous studies (Ono and Bono 2021; Bono and Hirota 2020). We then calculated scaledTPM for use this as a representative value for gene expression (Soneson et al. 2016). Our quantitative RNA-seq data are accessible at figshare (<https://doi.org/10.6084/m9.figshare.19679520.v1>)" (Lines 361-383)

Here a some additional minor comments:

Page 3, line 45, Content: "requires a move away"

Comment: Please rephrase: "moving away" or "to move away"

Response3-16: We corrected it as you suggested. (Line 39)

Page 5, line 76, Content: "overarching results for each chromosome."

Comment: Not sure what "overarching" means here

Response3-17: We admit that "overarching" was not appropriate. This sentence was removed because it corresponded to the result.

Page 5, line 82, Content: "Our data describe various analyses"

Comment: Not sure what this means, please rephrase

Response3-18: We rewrote the corresponding sentence due to inappropriate wording."

The analyses were designed to evaluate hypoxia-responsive genes and transcripts based on two reference transcript sets (Fig. 1). "(Lines 79-80)

Page 5, line 93, Content: "data set"

Comment: Use "dataset"

Reponse3-19: We corrected it as you suggested. (Line 89)

Page 9, lines 145-146, Content: " (A) Procedure for calculating HNF-score using FANTOM-CAT as a reference. "

Comment: I am not sure that presenting the alignment procedure here and in this form is useful. Panel A does not provide much useful information , this information should be in the methods.

Response3-20: Thanks for the suggestion. We removed the figure and put the information in methods section.

September 23, 2022

RE: Life Science Alliance Manuscript #LSA-2022-01518-TR

Prof. Hidemasa Bono
Hiroshima University
3-10-23 Kagamiyama
Higashi-Hiroshima, Hiroshima 739-0046
Japan

Dear Dr. Bono,

Thank you for submitting your revised manuscript entitled "Exploratory Meta-Analysis of Hypoxic Transcriptomes Using Precise Transcript Reference Sequence Set". We would be happy to publish your paper in Life Science Alliance pending final revisions necessary to meet our formatting guidelines.

-please add the Twitter handle of your host institute/organization as well as your own or/and one of the authors in our system

A. FINAL FILES:

B. MANUSCRIPT ORGANIZATION AND FORMATTING:

Sincerely,

Reviewer #1 (Comments to the Authors (Required)):

The authors revised the manuscript, and my original concerns were properly addressed in it. Thus, I would recommend it for publication.

September 26, 2022

RE: Life Science Alliance Manuscript #LSA-2022-01518-TRR

Prof. Hidemasa Bono
Hiroshima University
3-10-23 Kagamiyama
Higashi-Hiroshima, Hiroshima 739-0046
Japan

Dear Dr. Bono,

Thank you for submitting your Research Article entitled "Exploratory Meta-Analysis of Hypoxic Transcriptomes Using Precise Transcript Reference Sequence Set". It is a pleasure to let you know that your manuscript is now accepted for publication in Life Science Alliance. Congratulations on this interesting work.

DISTRIBUTION OF MATERIALS:

Again, congratulations on a very nice paper. I hope you found the review process to be constructive and are pleased with how the manuscript was handled editorially. We look forward to future exciting submissions from your lab.

Sincerely,
